# ATTRIBUTES RECONSTRUCTION IN HETEROGENEOUS NETWORKS VIA GRAPH AUGMENTATION

## ABSTRACT

Heterogeneous Graph Neural Networks(HGNNs), as an effective tool for mining heterogeneous graphs, have achieved remarkable performance on node classification tasks. Yet, HGNNs are limited in their mining power as they require all nodes to have complete and reliable attributes. It is usually unrealistic since the attributes of many nodes in reality are inevitably missing or defective. Existing methods usually take imputation schemes to complete missing attributes, in which topology information is ignored, leading to suboptimal performance. And some graph augmentation techniques have improved the quality of attributes, while few of them are designed for heterogeneous graphs. In this work, we study the data augmentation on heterogeneous graphs, tackling the missing and defective attributes simultaneously, and propose a novel generic architecture—Attributes Reconstruction in Heterogeneous networks via Graph Augmentation(ARHGA), including random sampling, attribute augmentation and consistency training. In graph augmentation, to ensure attributes plausible and accurate, the attention mechanism is adopted to reconstruct attributes under the guidance of the topological relationship between nodes. Our proposed architecture can be easily combined with any GNN-based heterogeneous model, and improves the performance. Extensive experiments on three benchmark datasets demonstrate the superior performance of ARHGA over strate-of-the-art baselines on semi-supervised node classification.

## 1 INTRODUCTION

Heterogeneous information networks(HINs)(Yang et al. (2020); Shi et al. (2016); Shen et al. (2017)), which contain multiple types of nodes and edges, have been widely used to model complex systems and solve practical problems. Recently, heterogeneous graph neural networks have emerged as prevalent deep learning architectures to analyze HINs and shown superior performance in various graph analytical tasks, such as node classification(Wang et al. (2019a); Yun et al. (2019)) and link prediction(Fu et al. (2020); Zhang et al. (2019)). Most HGNNs follow a message-passing scheme in which each node updates its embedding by aggregating information of its neighbors' attributes. Such message-passing scheme usually requires that all nodes have complete and reliable attributes, which is not always satisfied in practice due to resource limitation and personal privacy, resulting in missing and defective attributes.

In general, the attribute missing in heterogeneous graphs means that attributes of partial nodes are entirely missing, compared to that in homogeneous graphs, is more frequent and complex. Take DBLP(Sun & Han (2013)) as an example, the network has four types of nodes(author, paper, term and venue) and three types of links. Only paper nodes have attributes which are extracted from the keywords in their titles, while other types of nodes have no attributes. It impairs the effectiveness of the corresponding graph mining model to certain extents. In another fold, the original attributes of nodes are sometimes not ideal since heterogeneous graphs are extracted from complex systems which inevitably are subject to various forms of contamination, such as mistakes and adversarial attacks, making error propagation and greatly affecting the process of message-passing. This suggests the need for effective approaches able to complete missing attributes and calibrate defective attributes in heterogeneous graphs simultaneously.

To alleviate the effect incurred from missing attributes, the existing models usually adopt imputation strategy, such as neighbor's average or one-hot vector as done in MAGNN(Fu et al. (2020)). These

imputation methods are nonoptimal because graph structure information is ignored and only rare useful information is provided, hampering subsequent analysis. An alternative technique to tackle this issue is to consider graph topology information and inject it into the completion models. The work of Jin et al. (2021) and He et al. (2022) has shown a significant boost on node classification tasks. But both methods naturally assume that the original attributes are reliable, which is not easy to satisfy in real-world applications. In another concern of research, some graph augmentation techniques are adopted to calibrate original attributes to improve the quality and have shown a promising performance(Xu et al. (2022); Zhu et al. (2021)). However, these methods are deficient for heterogeneous graphs as they are not capable of encoding complex interaction. Further, existing methods either only complete missing attributes or only improve the quality of attributes, while it is worthy making efforts to solving both problems at the same time.

In this paper, we attempt to deal with the missing and defective attributes simultaneously in heterogeneous graphs, and propose a novel framework for Attributes Reconstruction in Heterogeneous networks via Graph Augmentation(ARHGA). ARHGA repeatedly sample nodes to perform attribute augmentation to obtain multiple augmented attributes for each node, and then utilize consistency training(Xie et al. (2020)) to make the outputs of different augmentations as similar as possible. Moreover, to ensure the augmented attributes more accurate, node topological embeddings are learned through HIN-embedding methods(Dong et al. (2017); Fu et al. (2017); Shang et al. (2016); Wang et al. (2019b)) to capture graph structure information as guidance. In this way, ARHGA effectively enhances the performance of existing GNN-based heterogeneous models in aid of the reconstructed attributes.

**Contributions.** In summary, the main contributions of this paper are as follows:

- We propose a generic architecture of graph augmentation on heterogeneous networks for attributes reconstruction, focusing both on the missing and defective attributes.
- We design an effective attribute-wise augmentation strategy implemented by attention mechanism, which integrates topology information to increase the reliability of the reconstructed attributes.
- Extensive experimental results on three node classification benchmark datasets demonstrate the effectiveness of our proposed model.

## 2 RELATED WORK

**Heterogeneous graph neural networks.** Heterogeneous graphs have been widely used to solve real-world problems due to a diversity of node types and relationships between nodes. Recently, many HGNNs have been proposed to analyze HINs. HAN(Wang et al. (2019a)) learns node representations using the node-level attention and the meta-path-level attention. Fu et al. (2020) further consider the intermediate nodes and propose the node-type specific transformation, then use a hierarchical attention structure similar to HAN. Graph Transformer Networks(Yun et al. (2019)) generates a new graph by identifying useful connections between unconnected nodes on the original graph, then performs graph convolution on the new graph. Zhang et al. (2019) sample heterogeneous neighbors for each node by random walk and aggregate node and type level information. Hu et al. (2020) design meta-relation-based mutual attention to handle graph heterogeneity and implicitly learn meta paths. Nevertheless, these methods require that all nodes have complete and reliable attributes and encounter difficulties in dealing with heterogeneous graphs with the missing and defective attributes.

**Learning with missing attributes.** Previous handcrafted approaches to fill in missing attributes rarely consider graph topology information, making inefficient attributes and compromising model performance. Several deep learning models have been explored. You et al. (2020) design a unified framework in which attribute completion is modeled as an edge-level prediction task and the label prediction as a node-level prediction task. Chen et al. (2020) develop a novel GNN framework to perform the link prediction task and the attribute completion task based on distribution matching. Despite tremendous success, they are not suitable for HINs with missing attributes because the complex interaction has not been addressed. Recent advance in HGNNs has provided new directions to solve the problem. Jin et al. (2021) attempt to complete missing attributes in a learnable manner and propose the framework HGNN-AC, which contains pre-training of topology embedding and attribute completion with attention mechanism. He et al. (2022) design an unsupervised heterogeneous graph contrastive learning approach for analyzing HINs with missing attributes. Both

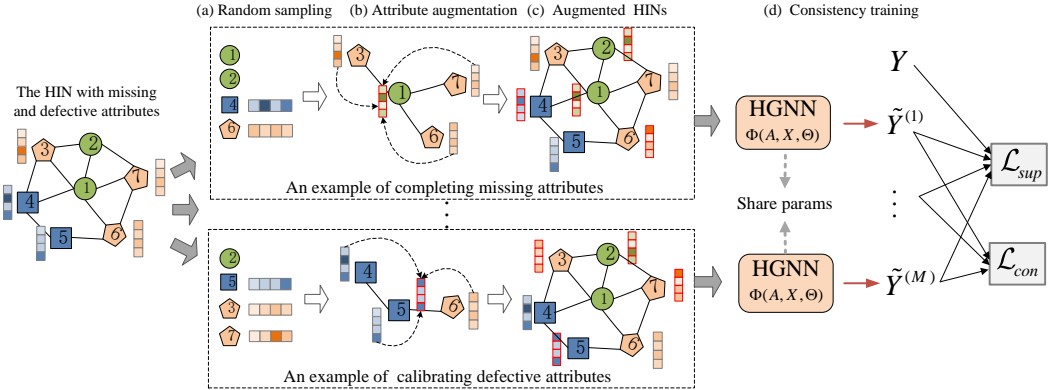

Figure 1: The overview of ARHGA framework. ARHGA performs random sampling (a) to produce the $ANS$. The attributes of nodes in the $ANS$ are completed or calibrated through attribute augmentation (b) to generate multiple graph augmentations (c), which are fed into to shared HGNNs to construct the consistency loss (d). The upper box shows an example of missing attributes in a sampling and the lower represents an example of defective attributes in another sampling.

methods only complete the missing attributes, while our proposed method not only completes the missing attributes but also calibrates defective attributes to enhance the quality of attributes.

**Graph data augmentation.** Recently, variants of data augmentation techniques have been explored in deep graph learning and demonstrated remarkable results. Graph data augmentation aims to generate the augmented graph(s) through enriching or changing the information from the original graph. It can be categorized into three classes: attribute-wise augmentation, structure-wise augmentation and label-wise augmentation(Ding et al. (2022)). Among the attribute-wise augmentation strategies, there are a few lines of existing works for attribute calibration/denoising. A straightforward method(Xu et al. (2022)) is to compute the gradient of specific objective functions w.r.t. the node attribute matrix and calibrate the node attributes matrix based on the computed gradient. In addition, as a special case of noisy setting on graph data, the problem of missing attributes also has been studied. Representative works include GCNMF and Feature Propagation. GCNMF(Taguchi et al. (2021)) completes the missing data by Gaussian Mixture Model(GMM). It integrates graph representation learning with attribute completion and can be trained in end-to-end manner. Feature Propagation(Rossi et al. (2021)) reconstructs the missing node attributes based on a diffusion-type differential equation on the graph. It is important to note that the above-mentioned graph augmentation methods only work on homogeneous graphs, heterogeneous attribute-wise augmentation is still an under-explored remains problem.

## 3 METHODOLOGY

In this section, we introduce the ARHGA framework, of which the general idea is to tackle the missing and defective attributes simultaneously through an effective graph augmentation strategy, and integrate the augmentation process and HGNN module into a unified framework to benefit the performance on node classification tasks. Figure 1 illustrates the proposed framework.

### 3.1 RANDOM SAMPLING

Random sampling randomly selects a part of nodes to form the augmentation node set($ANS$) in which nodes may have no attributes or have original attributes. Specifically, given a heterogeneous graph $\mathcal{G}$ with node set $\mathcal{V}$ of $n$ nodes, edge set $\mathcal{E}$ and attribute matrix $X$, we randomly sample a binary mask $\varepsilon_i \sim Bernoulli(1 - \delta)$ for each node $v_i$ in $\mathcal{V}$ to determine whether $v_i$ belongs to the $ANS$, where $\delta$ is the probability when $\varepsilon_i$ takes 0. We obtain $ANS = \{v_i | \varepsilon_i = 0\}$, i.e., if $\varepsilon_i = 0$, then $v_i \in ANS$. The attributes of nodes in the $ANS$ are dropped in this step and reconstructed in the phase of attribute augmentation.

To reconstruct attributes for all nodes, random sampling is performed repeatedly to guarantee that all nodes are selected. If the number of samplings is set as $M$, we give the constraint under which $M$ should satisfy from a probabilistic view. Let $P$ denote the probability of all nodes being selected at least once, since each node is sampled independently, we have:

$$P = \prod_{i=1}^{n} P_i = (P_i)^n, \tag{1}$$

$$P_i = 1 - P_i^C = 1 - P(k = 0) = 1 - \delta^M, \tag{2}$$

where $P_i$ and $P_i^C$ are the probability that node $v_i$ is selected at least once and that is not selected respectively, $k$ is the number of times being selected. Given a threshold $\tau$ close enough to 1, when $P$ is greater than $\tau$, it is considered having the full coverage of all nodes after $M$ random samplings. Substituting Eq.(2) into Eq.(1), it can be deduced that $M$ should satisfy $(1 - \delta^M)^n > \tau$, i.e., $M > \log_\delta(1 - \sqrt[n]{\tau})$.

Note that the sampling procedure is only performed during training. During inference, we directly consider the node set $\mathcal{V}$ as the $ANS$.

## 3.2 ATRRIBUTE AUGMENTATION

In heterogeneous graphs, attribute information and structure information are two crucial characteristics, and are semantically similar to each other due to the homophily of network(McPherson et al. (2001); Pei et al. (2020); Schlichtkrull et al. (2018)). With this principle, there is a reasonable hypothesis that the topological relationship between nodes can well reflect the relationship of nodes'attributes. Or, briefly, adjacent nodes tend to have similar attributes. So attributes of nodes in the $ANS$ can be reconstructed based on attribute information of their neighbors. Considering the complex interactions in heterogeneous networks, the node topological embeddings $H$ is learned to capture underlying structure information, and used as guidance to reconstruct attributes.

**Topological embedding.** In graph $\mathcal{G}$, the node set $\mathcal{V}$ is associated with corresponding node-type set $\mathcal{F}$ and the edge set $\mathcal{E}$ is associated with corresponding edge-type set $\mathcal{R}$. To capture the structure information in $\mathcal{G}$, random walk(Huang et al. (2019)) is adopted based on multiple common meta-paths, to generate more comprehensive node sequences which are fed into a heterogeneous skip-gram model to learn node topological embeddings. Given a pre-defined meta-path $\mathcal{P} : F_1 \xrightarrow{R_1} F_2 \xrightarrow{R_2} \cdots \xrightarrow{R_{l-1}} F_l$, for node $v_i$ with type $F_i$, the transition probability at step $i$ can be formulated as:

$$p(v_{i+1}|v_i, \mathcal{P}) = \begin{cases} \frac{1}{|\mathcal{N}_{F_{i+1}}(v_i)|}, & (v^{i+1}, v^i) \in \mathcal{E}, \phi(v_{i+1}) = F_{i+1} \\ 0, & otherwise \end{cases}, \tag{3}$$

where $N_{F_{i+1}}(v_i)$ stands for the neighbors with type $F_{i+1}$ of node $v_i$. The random walk makes that underlying semantics information in graph $\mathcal{G}$ is preserved properly. Then skip-gram(Mikolov et al. (2013)) with one-hot vector as input is adopted to learn topological embeddings by maximizing the probability of the local neighbor structures captured by the random walk, that is:

$$\max_{\theta} \sum_{v \in \mathcal{V}} \sum_{F \in \mathcal{F}} \sum_{u \in \mathcal{N}_F(v)} \log p(u|v; \theta), \tag{4}$$

where $\mathcal{N}_F(v)$ the set of neighbors with type $F$ of node $v_i$, which is sampled by random walk based on the given metapath. The topological embedding of nodes can be learned and denoted as $H$.

**Attribute-wise augmentation.** From the perspective of network science, the information on direct links is more essential, so one-hop neighbors have more contributions to attribute augmentation. And we notice that directly connected neighbors have different importance because the neighbors of a node may be of different types and different local structure in heterogeneous graphs. ARHGA adopts the attention mechanism to learn the importance of different one-hop neighbors based on topological embedding $H$. Note that ARHGA only computes attention coefficients of nodes' direct neighbors by performing masked attention, which reduces unnecessary computation and is thus more efficient.

Specifically, given a node pair $(v, u)$ which are directly connected, the attention layer learns the importance $e_{vu}$ which indicates the contribution of node $u$ to node $v$, $u \in \mathcal{N}_v^+$, where $\mathcal{N}_v^+$ denotes

the one-hop attributed neighbors of node $v$, the importance can be computed as:

$$e_{vu} = \sigma(h_v^{\mathrm{T}} W h_u), \tag{5}$$

where $h_u$ and $h_v$ are the topological embeddings of node $u$ and node $v$. Here the attention layer, parametrized by a weight matrix $W$, is shared for all node pairs, and $\sigma$ is an activation function. To better compare the importance across different nodes, the softmax function is used to normalize them to get weighted coefficient $\alpha_{uv}$:

$$\alpha_{vu} = softmax(e_{vu}) = \frac{\exp(e_{vu})}{\sum_{s \in \mathcal{N}_v^+} \exp(e_{vs})}. \tag{6}$$

Then, ARHGA obtains attributes of node $v$ by aggregating attributes of it's one-hop neighbors according to weighted coefficients $\alpha_{vu}$:

$$\tilde{x}_v = \sum_{u \in \mathcal{N}_v^+} \alpha_{vu} x_u. \tag{7}$$

In this way, the missing attributes are completed and original defective attributes are calibrated. As shown in Figure 2, we give a brief explanation on the author node(A1) in DBLP and the paper node(P1) in ACM as an example. The node A1 has no attributes and the node P1 has original defective attributes, their attributes are reconstructed effectively using our proposed framework. Importantly, the completed attributes are confirmed and the calibrated attributes are more meaningful than original attributes owing to integration with underlying semantic information in heterogeneous graphs after conducting weighted aggregation.

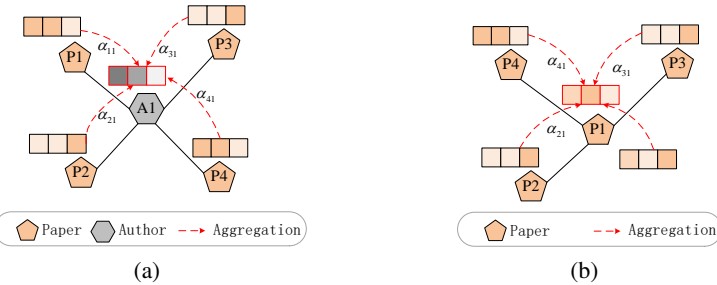

Figure 2: (a) An illustration of completing the attributes of node A1 in DBLP dataset. (b) An illustration of calibrating the attributes of node P1 in ACM dataset.

Finally, to stabilize the learning process and reduce the high variance (brought by the heterogeneity of graphs), we extend the attention mechanism to multi-head attention, as done in many existing methods(Veličković et al. (2017), Wang et al. (2019a)). Specifically, $K$ independent attention mechanisms are performed to serve as the final attributes of node $v$, and then their outputs are averaged to generate the following attributes of node $v$:

$$\tilde{x}_v = mean(\sum_k^K \sum_{u \in \mathcal{N}_v^+} \alpha_{vu} x_u), \tag{8}$$

where $mean(\cdot)$ represents an average function and $K$ is time of performing attention mechanism. After attribute-wise augmentation, attributes of all nodes are updated as:

$$\tilde{X} = \{\tilde{X}_i, X_j | v_i \in ANS, v_j \in V - ANS\}. \tag{9}$$

### 3.3 CONSISTENCY TRAINING

After performing random sampling and attribute augmentation for $M$ times, we generate $M$ augmented attribute matrices $\{\tilde{X}^{(m)} | 1 \leq m \leq M\}$, each of which together with the original graph structure $A$ is sent into the HINs model to get the corresponding output:

$$\tilde{Y}^{(m)} = \Phi(A, \tilde{X}^{(m)}), \tag{10}$$

where $\tilde{Y}^{(m)}$ are the prediction probabilities of $\tilde{X}^{(m)}$, $\Phi$ denotes a HINs model. With the augmented attributes, ARHGA can be applied to any other heterogeneous graph models and successfully enhance the performance. To be specific, MAGNN(Fu et al. (2020)) is combined when implementing ARHGA.

**Supervised Loss.** Under a semi-supervised setting, with $s$ labeled nodes among $n$ nodes, the supervised loss on the node classification task is defined as the average cross-entropy loss over $M$ augmentations:

$$\mathcal{L}_{sup} = -\frac{1}{M} \sum_{m=1}^{M} \sum_{i=0}^{s-1} Y_i^{\mathrm{T}} \log \tilde{Y}_i^{(m)}, \tag{11}$$

where $Y \in \{0, 1\}^{n \times C}$ are ground-truth labels with $C$ representing the number of classes.

**Consistency Loss.** Valid graph data augmentation changes the input in a way that has relatively trivial impact on the final node classification. We naturally embed this knowledge into our model by designing a consistency loss over $M$ augmentations.

For each node $v_i$, we generate a "guessed label" no matter it originally has a label or not. Specifically, find the average of the model's prediction distributions across all the $M$ augmentations of $v_i$:

$$\bar{Y}_i = \frac{1}{M} \sum_{m=1}^{M} \tilde{Y}_i^{(m)}, \tag{12}$$

then the "guessed label" $\bar{Y}_i' = (\bar{Y}_{i0}', ..., \bar{Y}_{ij}', \bar{Y}_{iC-1}')^{\mathrm{T}}$ for node $v_i$ is computed through the sharpening trick(Berthelot et al. (2019)), in which $\bar{Y}_{ij}'$ is the guessed probability of $v_i$ belonging to class $j$:

$$\bar{Y}_{ij}' = \bar{Y}_{ij}^{\frac{1}{T}} / \sum_{c=0}^{C-1} \bar{Y}_{ic}^{\frac{1}{T}} , 0 \leq j \leq C-1, \tag{13}$$

where $0 \leq T \leq 1$ acts as the "temperature" that controls the sharpness of the distribution. In ARHGA, $T$ is set as a small value to enforce the "guessed label" to approach a one-hot distribution. Then we minimize the distance between $\tilde{Y}_i$ and $\bar{Y}_i'$ to design the consistency loss:

$$\mathcal{L}_{con} = \frac{1}{M} \sum_{m=1}^{M} \sum_{i=0}^{n-1} ||\bar{Y}_i' - \tilde{Y}_i^{(m)}||_2^2. \tag{14}$$

**Optimization Objective.** Both the supervised loss and the consistency loss are combined as the final loss of ARHGA:

$$\mathcal{L} = \mathcal{L}_{\mathrm{sup}} + \lambda \mathcal{L}_{con}, \tag{15}$$

where the hyper-parameter $\lambda$ is introduced to control the balance between the two losses. By minimizing the final loss, our model can be optimized via back propagation in an end-to-end manner. Algorithm 1(present in Appendix A.1) outlines ARHGA's training process. During inference, we directly take $\mathcal{V}$ as the $ANS$, that is, attributes of all nodes are reconstructed after performing attribute augmentation one time. Hence, the inference formula is $\tilde{Y} = \Phi(A, \tilde{X}, \Theta)$, where $\tilde{X} = \{\tilde{X}_i | v_i \in \mathcal{V}\}$ and $\tilde{Y}$ is the corresponding prediction probabilities.

## 4 EXPERIMENTS

In this section, we first give the experimental setup, and then evaluate the performance of ARHGA on the node classification and report the visualization results. We also conduct a deep analysis on the effectiveness of ARHGA. Finally, we investigate the sensitivity with respect to hyper-parameters of ARHGA.

**Datasets.** We conduct experiments on three widely-used HINs datasets, i.e., DBLP[1], ACM[2], IMDB[3], to analyze the effectiveness of ARHGA. Note that, only paper nodes in DBLP and ACM,

---

[1]https://dblp.uni-trier.de/

[2]https://dl.acm.org/

[3]https://www.imdb.com/

Table 1: Results(%) of node classification on three datasets

| Datasets | Metrics | Training | metapath2vec | GCN | GAT | HetGNN | HAN | MAGNN | HGNN-AC | ARHGA |
|---|---|---|---|---|---|---|---|---|---|---|
| DBLP | Macro-F1 | 10% | 91.09 | 89.53 | 64.57 | 91.09 | 91.24 | 92.44 | 93.80 | **93.86** |
| | | 20% | 91.50 | 90.06 | 66.92 | 91.72 | 91.69 | 92.53 | 93.92 | **94.12** |
| | | 40% | 92.55 | 90.37 | 73.23 | 92.03 | 91.84 | 92.97 | 94.06 | **94.44** |
| | | 60% | 93.25 | 90.57 | 77.17 | 92.26 | 92.01 | 93.30 | 94.04 | **94.52** |
| | | 80% | 93.48 | 90.74 | 78.20 | 92.39 | 92.15 | 93.77 | 94.22 | **94.81** |
| | Micro-F1 | 10% | 91.74 | 90.02 | 75.90 | 91.64 | 91.88 | 93.02 | 94.22 | **94.31** |
| | | 20% | 92.14 | 90.53 | 76.98 | 92.23 | 92.30 | 93.08 | 94.34 | **94.55** |
| | | 40% | 93.09 | 90.83 | 79.61 | 92.55 | 92.46 | 93.50 | 94.46 | **94.84** |
| | | 60% | 93.76 | 91.01 | 81.62 | 92.79 | 92.65 | 93.83 | 94.46 | **94.91** |
| | | 80% | 93.94 | 91.15 | 82.22 | 92.92 | 92.78 | 94.27 | 94.61 | **95.17** |
| ACM | Macro-F1 | 10% | 67.81 | 70.79 | 89.29 | 88.20 | 89.39 | 86.60 | 90.29 | **92.07** |
| | | 20% | 69.95 | 70.41 | 89.59 | 89.16 | 90.01 | 88.01 | 91.51 | **92.94** |
| | | 40% | 71.15 | 70.82 | 89.77 | 90.14 | 90.82 | 89.42 | 92.75 | **94.02** |
| | | 60% | 71.74 | 69.67 | 89.72 | 90.71 | 91.51 | 90.39 | 93.46 | **94.36** |
| | | 80% | 72.18 | 67.23 | 89.42 | 91.01 | 91.71 | 90.79 | 93.79 | **94.57** |
| | Micro-F1 | 10% | 70.29 | 74.10 | 89.19 | 88.18 | 89.32 | 86.67 | 90.43 | **91.95** |
| | | 20% | 72.12 | 74.02 | 89.47 | 89.12 | 89.89 | 88.08 | 91.64 | **92.82** |
| | | 40% | 73.17 | 74.57 | 89.65 | 90.11 | 90.73 | 89.48 | 92.9 | **93.94** |
| | | 60% | 73.65 | 74.10 | 89.60 | 90.64 | 91.37 | 90.42 | 93.57 | **94.29** |
| | | 80% | 74.14 | 72.69 | 89.29 | 90.93 | 91.56 | 90.80 | 93.87 | **94.52** |
| IMDB | Macro-F1 | 10% | 44.29 | 43.70 | 53.61 | 45.68 | 57.02 | 56.39 | 58.69 | **59.39** |
| | | 20% | 46.42 | 44.75 | 54.81 | 48.92 | 57.61 | 58.11 | 59.67 | **60.45** |
| | | 40% | 47.70 | 45.26 | 55.09 | 51.61 | 57.75 | 59.39 | 60.18 | **60.98** |
| | | 60% | 48.25 | 47.70 | 55.71 | 53.00 | 57.66 | 59.97 | 60.60 | **61.39** |
| | | 80% | 48.73 | 48.25 | 55.40 | 53.24 | 57.23 | 60.02 | 60.75 | **61.71** |
| | Micro-F1 | 10% | 46.15 | 47.02 | 54.14 | 46.56 | 57.35 | 56.53 | 58.97 | **59.60** |
| | | 20% | 48.08 | 47.44 | 55.02 | 49.70 | 57.82 | 58.16 | 59.84 | **60.53** |
| | | 40% | 49.55 | 47.62 | 55.29 | 52.47 | 57.98 | 59.46 | 60.38 | **61.09** |
| | | 60% | 50.06 | 48.49 | 55.91 | 53.91 | 57.87 | 60.05 | 60.79 | **61.52** |
| | | 80% | 50.68 | 48.73 | 55.67 | 54.25 | 57.46 | 60.15 | 60.98 | **61.88** |

and movie nodes in IMDB have original attributes while other types of nodes have on attributes. More details are present in Appendix A.2.1.

**Baselines.** We compare ARHGA with seven state-of-the-art methods representative of two different categories: two traditional homogeneous GNNs, i.e., GCN(Kipf & Welling (2016)), GAT(Veličković et al. (2017)), and five heterogeneous graph models, i.e., metapath2vec(Dong et al. (2017)), HetGNN(Zhang et al. (2019)), HAN(Wang et al. (2019a)), MAGNN(Fu et al. (2020)), MAGNN-AC(Jin et al. (2021)). Among heterogeneous graph models, metapath2vec is a network embedding methods and other models are GNN-based methods. MAGNN-AC is an approach to complete missing attributes of nodes having no attributes in heterogenerous graphs.

**Parameter Settings.** For the settings/parameters of different baselines, we use the default hyper-parameters suggested in MAGNN-AC. The embedding dimensions of all methods is set to 64 for a fair comparison. In ARHGA, random sampling probability $\delta$ is set to 0.5 and the number of augmentations $M$ to 6. The number of attention heads K is set to 8 and $\lambda$ is 0.5 for the weight of the consistency loss. In addition, ARHGA is trained by adopting Adam optimizer(Kingma & Ba (2014)) with the learning rate 0.005. The early stopping strategy with a patience of 5 epoches is adopted on validation set in our experiments.

## 4.1 NODE CLASSIFICATION

In this section, the performance on node classification of ARHGA is compared with that of different models. First, we generate embeddings of labeled nodes(i.e., authors in DBLP, authors in ACM and movies in IMDB), and then feed them into a linear support vector machine(SVM)(Suykens (2001)) classifier with different training ratios from 10% to 80%. Table 1 reports the results of averaged Macro-F1 and Micro-F1 over 5 times, where the best results are in bold fonts.

As shown, ARHGA consistently achieves a significant margin over baselines and datasets. After conducting attribute augmentation, ARHGA has 0.9%-5.47% higher accuracy than MAGNN and 0.06%-1.78% higher than MAGNN-AC, another MAGNN-based method, which performs the best results from the baseline methods. Specially, although some methods have already achieved high accuracy on DBLP dataset, ARHGA still has nontrivial improvement. The improvement is primarily due to the fact ARHGA provides a better representation of node attributes. In addition, GNN-based

heterogeneous methods perform better than metapath2vec since attributes information is integrated, demonstrating the importance of node attributes. The poor performance of GCN and GAT reveals the importance of encoding heterogeneous semantic information when analyzing HINs. Our proposed method not only captures and preserves underlying semantic information in heterogeneous graphs but also effectively reconstructs node attributes to augment graph, which contribute to the great superiority of model.

## 4.2 VISUALIZATION

For a more intuitive comparison, we conduct the task of visualization by learning embeddings of paper nodes of MAGNN, MAGNN-AC and ARHGA on the ACM dataset. Then the well-known t-SNE(Van der Maaten & Hinton (2008)) is utilized to project the embeddings into a 2-dimensional space, where nodes with different colors belong to different class.

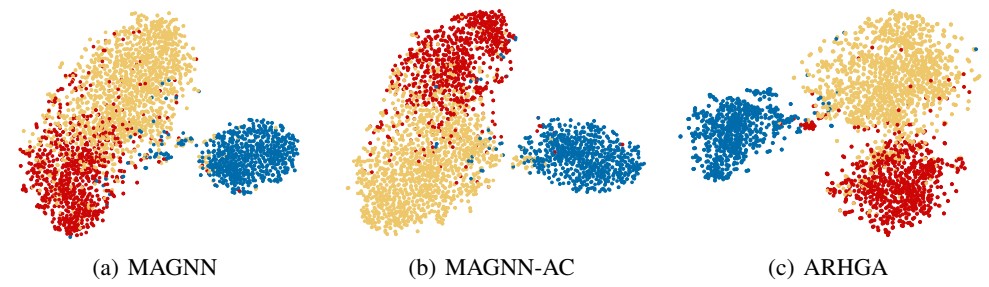

(a) MAGNN       (b) MAGNN-AC       (c) ARHGA

Figure 3: Visualization of embeddings of paper nodes in ACM. Different colors correspond to different research in ground truth.

As shown in Figure 3, with the reconstructed attributes, ARHGA has the clearest boundary and densest cluster structure to classify nodes among the three methods. In contrast, MAGNN and MAGNN-AC perform poorly, some paper nodes of different classes are mixed and overlapped obviously. MAGNN adopts imputation methods to fill in the missing attributes, which provides little useful information for node classification, resulting in poor performance. MAGNN-AC only completes the missing attributes and the defective attributes are not addressed properly. ARHGA not only completes missing attributes but also enhances the quality of attributes to make nodes more distinguishable.

## 4.3 A DEEP ANALYSIS OF ARHGA

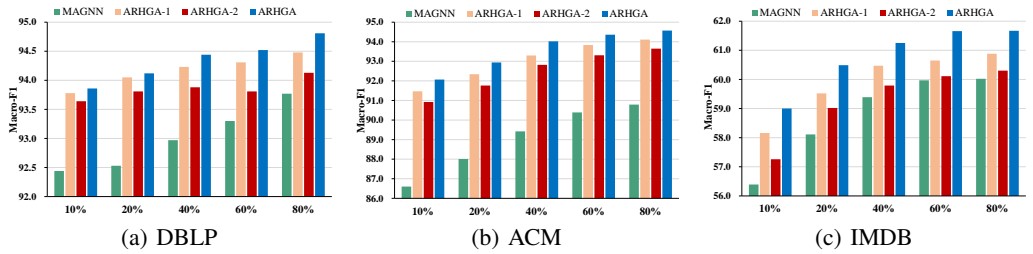

(a) DBLP        (b) ACM        (c) IMDB

Figure 4: Comparisons of ARHGA with two variants(ARHGA-1, ARHGA-2) and MAGNN on node classification.

To verify the effectiveness of attribute completion and attribute calibration in our method, we conduct experiments on comparing ARHGA with two variants. The variants are given as follows: 1) ARHGA of removing attributes calibration, only completing the missing attributes, named ARHGA-1; 2) ARHGA of removing attributes completion, only enhancing the quantity of the original defec-

tive attributes, named ARHGA-2, in which, the missing attributes are obtained by imputation strategies. In addition, we also compare with MAGNN to better analyze the benefit of ARHGA. Figure 4 presents the results of the deep analysis. More results see Appendix A.2.2.

As shown, ARHGA has the best results on all datasets at all label rates. Though ARHGA-1 and ARHGA-2 are lower than ARHGA, still outperform MAGNN. This is mainly because ARHGA-1 completes missing attributes accurately and ARHGA-2 calibrates the defective attributes effectively, while there are no corresponding strategies to tackle missing and defective attributes in MAGNN. Furthermore, all the results of ARHGA-1 are better than those of ARHGA-2, so it is believed that attribute completion contributes more than attribute calibration to the effectiveness of ARHGA. The overall results show that accurate attribute information palys a significant role in learning node representations. And ARHGA is an effective exploration of attribute reconstruction through graph data augmentation.

### 4.4 PARAMETER ANALYSIS

In this section, we investigate the sensitivity of ARHGA to the critical hyper-parameters: random sampling probability $\delta$, number of augmentations $M$ and weight of consistency loss $\lambda$. We report the results of node classification by varying these parameters on ACM dataset.

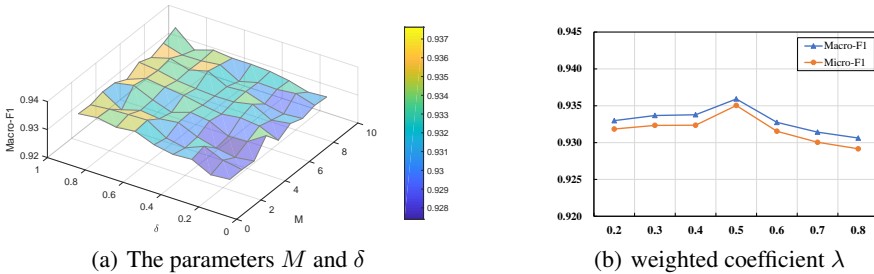

(a) The parameters $M$ and $\delta$        (b) weighted coefficient $\lambda$

Figure 5: Sensitivity analysis on ACM dataset. We report the average result of the node classification across different training ratios.

From Figure 5(a), the performance of ARHGA fluctuates slightly when $\delta$ and $M$ are in certain ranges, and drops obviously when $\delta$ and $M$ are set too small. This is because small $\delta$ and $M$ may result in attributes of some nodes not being reconstructed, impairing the effectiveness of graph augmentation. From Figure 5(b), we observe that the performance of ARHGA shows a trend of first rising and then slowly decreasing when $\lambda$ takes different values from 0.2 to 0.8, and achieves the best when $\lambda = 0.5$. This observation indicates that the consistency loss and supervised loss are both important, each of them contributes to the training of ARHGA. The overall results suggest that ARHGA is relatively stable for hyperparameter perturbation, demonstrating the insensitivity of ARHGA. The results on DBLP and IMDB dataset(provided in Appendix A.2.3) also indicate the similar sensitivity.

## 5 CONCLUSIONS

In this paper, we present a novel generic architecture(ARHGA) to reconstruct attributes in heterogeneous graphs. In ARHGA, we design an effective attribute augmentation strategy, which not only solves the problem of missing attributes, but also enhances the quality of original defective attributes. The augmentation strategy is essentially to conduct weighted aggregation through the attention mechanism guided by the topological relationship between nodes. The results of node classification show its consistent performance superiority over seven state-of-the-art baselines on benchmark datasets. The deep analysis of ARHGA demonstrates the effectiveness of completion of missing attributes and calibration of defective attributes. We conclude that ARHGA provides a better attribute representation helpful for improving semi-supervised classification on heterogeneous networks. In future work, we aim to design a structure-wise augmentation for a better graph structure representation in heterogeneous graphs.

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

# A    APPENDIX

## A.1    PSEUDOCODE OF ARHGA

---

**Algorithm 1** ARHGA

---

**Input:** The node set $\mathcal{V}$, adjacency matrix $A$, attribute matrix $X \in \mathbb{R}^{n \times d}$, time of augmentations $M$ in each epoch, the number of attention head $K$, learning rate $\eta$, a HIN model: $\Phi(A, X, \Theta)$.
**Output:** Prediction $\tilde{Y}$.
1: **while** not convergence **do**
2:    **for** $m = 1 : M$ **do**
3:        Randomly select nodes from $\mathcal{V}$ to compose the $ANS^{(m)}$
4:        Perform attribute augmentation on the $ANS^{(m)}$: $\tilde{x}_v^m = mean(\sum\limits_{k}^{K} \sum\limits_{u \in \mathcal{N}_v^+} \alpha_{vu} x_u)$
5:        Predict probability distribution using the HINs model: $\tilde{Y}^{(m)} = \Phi(A, \tilde{X}^{(m)}, \Theta)$
6:    **end for**
7:    Compute the supervised loss $\mathcal{L}_{sup} = -\frac{1}{M} \sum\limits_{m=1}^{M} \sum\limits_{i=0}^{s-1} Y_i^{\mathrm{T}} \log \tilde{Y}_i^{(m)}$
    and the consistency loss $\mathcal{L}_{con} = \frac{1}{M} \sum\limits_{m=1}^{M} \sum\limits_{i=0}^{n-1} ||\bar{Y}_i' - \tilde{Y}_i^{(m)}||_2^2$
8:    Update the parameters by gradients descending: $\Theta = \Theta - \eta \nabla_\Theta (\mathcal{L}_{sup} + \lambda \mathcal{L}_{con})$
9: **end while**
10: Output prediction $\tilde{Y}$ via: $\tilde{Y} = \Phi(A, \tilde{X}, \Theta)$

---

## A.2    MORE EXPERIMENTAL RESULTS

In this section, we provide dataset details and more experimental results besides the results in the main paper.

### A.2.1    ADDITIONAL DATASET DETAILS

In this section, we provide some additional, relevant dataset details, the statistics of these datasets are shown in Table A1.

Table A1: Statistics of the datasets

| Datasets | Nodes | Edges | HasAttributes |
|---|---|---|---|
| DBLP | Author(A):4057 Paper(P):14328 Term(T):7728 Venue(V):20 | A-P:19645 P-T:85810 P-V:14328 | Paper |
| ACM | Paper(P):4019 Author(A):7167 Subject(S):60 | P-P:9615 P-A:13407 P-S:4019 | Paper |
| IMDB | Movie(M):4278 Director(D):2081 Actor(A):5257 | M-D:4278 M-A:12828 | Movie |

- DBLP: This is a subset of DBLP with 4057 authors(A), 14328 papers(P), 20 venue(V), and 8789 terms(t). The authors are divided into four research areas: database, data mining, machinesearching, and information retrieval. Only paper nodes have attributes derived from their keywords, and other nodes have no raw attributes.

- ACM: This is a subset of DBLP with 4057 authors(A), 14328 papers(P), 20 venue(V), and 8789 terms(t). The authors are divided into four research areas: database, data mining, machinesearch-

ing, and information retrieval. Only paper nodes have attributes derived from their keywords, and other nodes have no original attributes.

- IMDB: We extract a subset of IMDB with 4278 movies(M), 2081 directors(D), and 5257 actors(A). The movie nodes are labeled by their genres: action, comedy and drama. In this dataset, movies are described by bag-of-words representation of their plot keywords, and other nodes have no original attributes.

### A.2.2 A DEEP ANALYSIS OF ARHGA

We also report the Micro-F1 values on three benchmark datasets to evaluate the effectiveness of attribute completion and attribute calibration in ARHGA. Figure 6 presents the results, which is similar to that of ACM dataset.

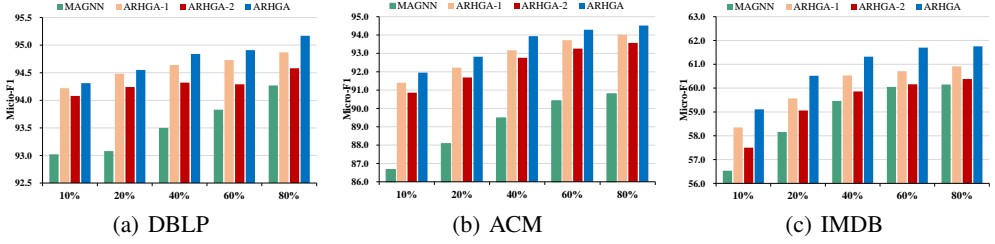

(a) DBLP       (b) ACM       (c) IMDB

Figure 6: Comparisons of ARHGA with two variants(ARHGA-1, ARHGA-2) and MAGNN on node classification.

### A.2.3 PARAMETER ANALYSIS

We investigate the sensitivity of critical hyper-parameters on DBLP and IMDB and report Macro-F1 values in Figure 7 and Figure 8. The results also demonstrate the insensitivity of ARHGA to hyperparameter perturbation.

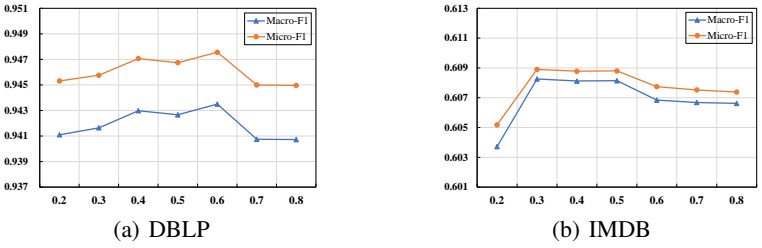

(a) DBLP       (b) IMDB

Figure 7: Analysis of the parameter sensitivity of $\lambda$. We report the average result of the node classification across different training ratios.

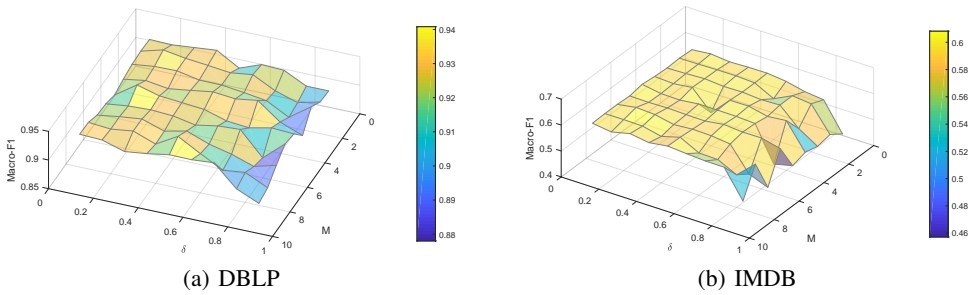

(a) DBLP       (b) IMDB

Figure 8: Analysis of the parameter sensitivity of $M$ and $\delta$. We report the average result of the node classification across different training ratios.

