# OpenReview forum: "ATTRIBUTES RECONSTRUCTION IN HETEROGENEOUS NETWORKS VIA GRAPH AUGMENTATION"
_ICLR.cc/2023/Conference — Submitted to ICLR 2023_

### Official Review · Reviewer_Aba8 · 2022-10-23

**Confidence:** 3
**Correctness:** 3
**Technical Novelty And Significance:** 2
**Empirical Novelty And Significance:** 3
**Recommendation:** 5

**Clarity, Quality, Novelty And Reproducibility:**

In my opinion, this paper is organized and written clearly. The author provides source codes and settings of experiments. The novelty of this paper is a little deficient.

**Strength And Weaknesses:**

Strength:

-- The attribute reconstruction problem in heterogeneous information networks is important. The topic of this paper is relative.

-- The proposed method is effective in some extent, which is proved by many experiments.

-- The method is introduced clearly and the paper is easy to follow.

Weakness:

-- The main contribution is a little incremental. In one hand, the framework is based on existing techniques. In the other hand, this paper does not provide any novel theoretical insights on this issue.

-- In my opinion, the experiments are not quite sufficient in following two perspectives. 1) The defective attributes are not covered in the experiments. One of the main contributions is to jointly tackle missing and defective attributes. The adopted datasets naturally have missing attributes, but there are no evidences that the attributes of these datasets are defective. It's better to change the original attributes randomly or exploit some simple adversarial attacks on attributes to make the original attributes defective. 2) There could be more downstream tasks other than node classification. Attribute reconstruction is widely used in many scenarios e.g. anomaly detection, unsupervised or self supervised learning. It's better to show that the proposed ARHGA framework could support different downstream tasks to prove the effectiveness of ARHGA.

-- Some typos and small writing problems, e.g. the mapping $\phi$ in Equation (3) is not explained in the paper.

**Summary Of The Paper:**

This paper proposed ARHGA, a heterogeneous graph augmentation method for attribute reconstruction. It solved both missing attributes and defective attributes problems in heterogeneous information networks. Extensive experiments prove the effectiveness of proposed attribute reconstruction framework.

**Summary Of The Review:**

This paper provides a graph augmentation method to solve the attribute reconstruction problem in HINs, however the contribution is incremental and the experiments could not fully support all the contributions.

---

### Official Review · Reviewer_Qaru · 2022-10-25

**Confidence:** 3
**Correctness:** 3
**Technical Novelty And Significance:** 1
**Empirical Novelty And Significance:** Not applicable
**Recommendation:** 1

**Clarity, Quality, Novelty And Reproducibility:**

The paper is well-written and easy to follow. The novelty is low as explained above.

**Strength And Weaknesses:**

The idea of dealing with missing/corrupted attributes in heterogeneous graphs is appealing. The authors proposed a method to tackle the problem by leveraging techniques like subgraph random sampling, graph embedding/attentions, and dual losses (supervised/consistency) is sound.

However, I do notice a similar method, HGNN-AC (WWW 21'). HGNN-AC is also cited in this work but not really thoroughly compared and discussed. Moreover, there is another similar method, (He et al., 2022). Both methods were only briefly mentioned in section 2. The only key difference that the author pointed out is they only complete missing attributes but do not fix defective attributes. Notably, both HGNN-AC and He et al., use almost an identical approach: random drop attributes, use topology embedding + attention, and dual losses (prediction loss + completion loss). The experiments also identically follow.

1) The authors need to better explain the novelty of this work and how it compares to HGNN-AC.
2) For experiments, only node classification was shown. Are there other possible tasks we can try?
3) The experiments are solely based on dropping attributes, how about attributes corruption/defective attributes?
4) How are "defective attributes" defined? Are there side effects of trying to update the attributes?
5) Could we also do a comparison with He et al., 2022 work? I understand it is unsupervised but the results in table 3 from He's work looks similar to ARHGA.

**Summary Of The Paper:**

The authors proposed a method, ARHGA, for inferring missing attributes and calibrating defective attributes in a heterogenous graph. The main idea of the paper is to subsample the graph and augment the attributes. The attributes were filled in using topological embeddings (based on random walk 2 vec on meta-paths) and a one-hope attention mechanism.



**Summary Of The Review:**

I recommend rejection due to a lack of novelty as explained above.

---

### Official Review · Reviewer_FtXi · 2022-11-01

**Confidence:** 4
**Correctness:** 3
**Technical Novelty And Significance:** 2
**Empirical Novelty And Significance:** 2
**Recommendation:** 3

**Clarity, Quality, Novelty And Reproducibility:**

The novelty of the paper is quite limited. The proposed method is a combination of (1) metapath2vec, (2) GAT, and (3) consistency loss.
The writings of the paper need to be polished and proofread. There exist lots of typos and grammar mistakes.

**Strength And Weaknesses:**

Strengths:
- The problem of heterogeneous graph augmentation is interesting and important.
- Experimental results can show some improvements.

Weaknesses:
- For Eq. (2), the authors claim that delta should be close enough to 1, but in the experiments, it's set to be 0.5 only
- The attention mechanism used in the paper is problematic. Attention-based aggregation is designed and applied to be among all types of nodes for attribute augmentation.
- The experimental results in Table 1 only shows very limited improvements over the baseline methods.
- Results are not reported by the mean and standard deviations.


**Summary Of The Paper:**

The paper proposes a joint learning on node classification and attribute augmentation on heterogeneous graphs. The attribute augmentation is based on the aggregations among one-hop neighbors weighted by attentions. Experiments show the proposed approach results in marginal improvements over the baseline methods.

**Summary Of The Review:**

Overall, the paper has limited novelty and contributions to be accepted:
- It's not clear how to set M based on delta. When delta is close to 1, M can be very large, which will be inefficient.
- The key idea of attention-based aggregation among one-hop neighbors (e.g., Eq. 7) is not meaningful. Different types of nodes have different input node attributes, thus directly aggregating x_u where u can be of any types does not make any sense. E.g., aggregating x_u where u is a paper for a venue node.
- The experimental results in Table 1 only shows very limited improvements over the baseline methods.
- Results are not reported by the mean and standard deviations.

---

### Decision · Program_Chairs · 2023-01-20

**Decision:**

Reject

**Justification For Why Not Higher Score:**

Reviewers too low.

**Justification For Why Not Lower Score:**

N/A

**Metareview: Summary, Strengths And Weaknesses:**

The title is very descriptive.   We have heterogeneous graphs and we want to impute missing values and repair defective values.  Graph augmentation is integrated with a GNN architecture to address the problem.

This is an important problem, the paper is clear and the experiments demonstrate the approach.

The experiments were missing some aspects ("defective" attributes, downstream evaluation), the method was incremental and some related work existed, HGNN-AC from WWW 21.

The authors did not respond to the reviewers, and the reviewers had a consensus on rejection.



**Summary Of Ac-Reviewer Meeting:**

N/A